# Traps and Pitfalls—Unspecific Reactions in Metabolic Engineering of Sesquiterpenoid Pathways

**DOI:** 10.3390/molecules25081935

**Published:** 2020-04-22

**Authors:** Maximilian Frey

**Affiliations:** Institute of Biology, Dept. of Biochemistry of Plant Secondary Metabolism (190b), University of Hohenheim, Garbenstraße 30, 70593 Stuttgart, Germany; maximilian_frey@uni-hohenheim.de

**Keywords:** rearrangement reactions, sesquiterpenes, sesquiterpene lactones, conjugation, metabolic engineering, enzyme characterization, transannular cyclization, Cope rearrangement

## Abstract

The characterization of plant enzymes by expression in prokaryotic and eukaryotic (yeast and plants) heterologous hosts has widely been used in recent decades to elucidate metabolic pathways in plant secondary metabolism. Yeast and plant systems provide the cellular environment of a eukaryotic cell and the subcellular compartmentalization necessary to facilitate enzyme function. The expression of candidate genes in these cell systems and the identification of the resulting products guide the way for the identification of enzymes with new functions. However, in many cases, the detected compounds are not the direct enzyme products but are caused by unspecific subsequent reactions. Even if the mechanisms for these unspecific reactions are in many cases widely reported, there is a lack of overview of potential reactions that may occur to provide a guideline for researchers working on the characterization of new enzymes. Here, an across-the-board summary of rearrangement reactions of sesquiterpenes in metabolic pathway engineering is presented. The different kinds of unspecific reactions as well as their chemical and cellular background are explained and strategies how to spot and how to avoid these unspecific reactions are given. Also, a systematic approach of classification of unspecific reactions is introduced. It is hoped that this mini-review will stimulate a discussion on how to systematically classify unspecific reactions in metabolic engineering and to expand this approach to other classes of plant secondary metabolites.

## 1. Introduction

The sesquiterpenes (ST) are a subgroup of the terpenes with a C15 backbone comprised of three isoprene units; sesquiterpenes with a lactone moiety are called sesquiterpene lactones (STL). The elucidation and metabolic engineering of their biosynthetic pathways have made significant progress in the past decades, as many sesquiterpenes are of commercial interest as fragrances [1], biodiesel [2] or pharmaceuticals [3,4]. The reconstruction of sesquiterpenoid metabolic pathways has been carried out in various prokaryotic and eukaryotic host models such as *Escherichia coli* [5,6], *Saccharomyces cervisiae* [4,7,8,9,10,11,12], *Nicotiana benthamiana* [7,8,13,14,15,16] and *Physcomitrella patens* [17,18]. The analysis of sesquiterpenes is usually performed by gas or liquid chromatography coupled with mass spectrometry, depending on the volatility of the compound. In the first step of terpene biosynthesis the carbon backbone is formed by a terpene synthase [19]. In a second step, the intermediate is often modified by cytochrome P450 enzymes that introduce oxygen into the core backbone [20,21]. Cytochrome P450 enzymes have been reported to introduce hydroxy-, epoxy-, acid- and estergroups [7,8,9,10,13] and the conversion from a germacrene to a guaiane backbone [12]. One of the challenges when expressing biosynthetic enzymes of sesquiterpenoid pathways is the differentiation of the direct enzyme product and artificial products that arise from unspecific subsequent reactions. Here, a concise overview of these unspecific reactions and how to avoid them is presented.

## 2. Unspecific Reactions

Four categories of unspecific reactions in pathway engineering of sesquiterpenoids can be classified: 1) S-conjugation, 2) O-conjugation, 3) acid-induced rearrangement and 4) heat-induced rearrangement. In each category several unspecific reactions can be observed and several combinations of these can occur. These unspecific reactions are each given a letter from (a) to (k). Irradiation is known to induce rearrangement reactions in germacranolide STL [22,23,24] as well. So far, there are no reports on unspecific reactions in metabolic engineering of sesquiterpenes caused by light irradiation, yet. However, we know that light does play an important role in the formation of artemisinin in nature [25]. Also, the influence of endogenous enzymes of the host cell system that convert the enzyme product is possible [26]. 

### 2.1. S-Conjugation

When expressing the genes of the metabolic pathway to costunolide from various species [7,15,26,27] in *Nicotiana benthamina*, the predominant product found was not free. Instead costunolide had mostly undergone (a) conjugation to cysteine (**1**) to form costunolide-cysteine (**2**) or (b) conjugation to glutathione to form costunolide-glutathione (**3**). The production of costunolide derivatives by expression of the corresponding metabolic pathway in *Nicotiana* such as 3β-costunolide, 14-hydroxycostunolide, eupatolide, parthenolide and 3β-parthenolide resulted in the formation of the cysteine and glutathione conjugates as well [8,13,14]. Also, during *in planta* production of inunolide, the 7,8-*cis* lactone isomer of costunolide resulted in cysteine and glutathione adducts [13]. When the same pathways were expressed in yeast, no cysteine or glutathione conjugates occurred [7,9,14,15].

### 2.2. O-Conjugation

The production of artemisinic acid (**4**) in *Nicotiana benthamiana* (Figure 1a) has been observed to yield mostly artemisinic acid 12-β-glucoside (**5**), which can be explained by (c) an esterification of the acid moiety of artemisinic acid to diglucose [28]. *In planta* produced epi-kunzeaol (**6**) was linked to two glucose units [29]. This was due to (d) an etherification of the C7-hydroxy moiety of epi-kunzeaol to form epi-kunzeaol-diglucoside (**7**).

### 2.3. Acid-Induced Rearrangement

Acidic conditions are known to induce transannular cyclization in germacrenes (Figure 1b), which can lead to a great number of rearrangement products, mostly with the C10 ring of a germacrene cyclizing to two C6 rings [22]. In the case of inunolide (**8**) a rearrangement product seems possible with (e) the double bond flipping to the C5-C6 position to form alantolactone (**9**) [13]. One well-observed example is as the rearrangement reaction from a germacrene to a eudesmane backbone. This acid-induced rearrangement converts, for instance, germacrene A acid (**10**) to α-costic acid (**11**), β-costic acid (**12**), and γ-costic acid (**13**) [9,10,30] with the double bond positions Δ3→4 (f), Δ4→15 (g) or Δ4→5 (h). The subsequent introduction of water (i) resulting in ilicic acid (**14**) has also been reported [9,10], likely neutralizing a carbocationic intermediate.

### 2.4. Heat-Induced Rearrangement

When analyzing the enzyme products of germacrene A synthases from various species in yeast and *Nicotiana* by GC-MS (Figure 1b) the enzyme product germacrene A (**15**) had converted to β-elemene (**16**) by Cope rearrangement [11,30]. Generally, the Cope rearrangement describes the heat-induced cyclization of 1,5-dienes [31].

## 3. Where Do the Reactions Happen?

S-conjugation and O-conjugation have so far only been observed in heterologous expression in plant cell systems, where enzyme products accumulate intracellularly (Figure 1c). Acid-induced rearrangement reactions have been observed in plant cell systems [13], during yeast cultivation in unbuffered yeast media and during extraction of yeast cultures [9,10] as well as during liquid and gas chromatography [10,32]. Heat-induced rearrangement has been observed in GC-MS applications [10,11,13,33].

## 4. What Are the Underlying Mechanisms and Avoiding Strategies?

### 4.1. S-Conjugation

S-conjugation to the thiol group of cysteine and O-conjugation to the hemiacetal group of glucose are mainly interpreted as a detoxifying mechanism of the host plant cell [7]. Formation of conjugates and transport to the vacuole may allow the plant cells of *Nicotiana benthamiana* to tolerate otherwise toxic cellular concentrations of bioactive metabolites, such as the STL. These conjugates are not observed in yeast expression systems where the enzyme products leave the cells by so far unknown mechanisms and accumulate in the culture medium [20]. The involvement of a glutathione-*S-*transferase (GST) or a non-enzymatic Michael-type addition have been suggested as an underlying mechanism [7] for cysteine (cys) or glutathione (GSH) conjugates (Table 1). Interestingly, only STL with an exocyclic α-methylene-γ-lactone group have so far been shown to form cys and GSH adducts to the lactone. Examples for this are costunolide (**1**), inunolide (**8**) and its derivatives (exocyclic methylene group: double bond position Δ11→13). On the other hand, the *inplanta* production of epi-dihydrocostunolide (**6**), an STL without an exocyclic methylene group, did not result in cysteine or GSH adducts, which would support a nonenzymatic Michael-type addition [29]. This nucleophilic addition reaction of an α-methylene-γ-lactone with the thiol group of biomolecules such as cysteine has been known for a long time [34] as one of the main reasons for STL bioactivity. It constitutes a special type of Michael addition [35] with an unsaturated lactone in which the nucleophile is a thiol group instead of an enolate [36]. Some STL, such as helenalin can also undergo Michael-type addition via their cyclopentenone moiety with an α,β-unsaturated carbonyl group [37,38]. Consequently, when expression of a metabolic pathway in *Nicotiana* leading to a STL with cyclopentenone moiety is performed, an additional cys or GSH adduct could be expected. GST-tagged STL may be transported into the vacuole by specific transporters and accumulate in the vacuole [7]. The cysteine conjugates are interpreted as breakdown products of the glutathione adducts or result from free cysteine inside the cells that reacts with the STL [7].

### 4.2. O-Conjugation

Sesquiterpenoids can form di-glucose adducts via esterification of an acid group or, in some cases, etherification of a hydroxy group to the hemiacetal group of glucose. While the formation of S-conjugates could be nonenzymatic due to the spontaneous Michael-type reaction at room temperature the introduction of two glucose units is most likely due to endogenous glycosyl transferases, which lead to the transport of the diglucose-“tagged” sesquiterpene to the vacuole [28]. Interestingly, no glycosylation of artemisinic acid (**4**) was observed when Fuentes et al. (2016) expressed the metabolic pathway to artemisinic acid in the chloroplasts of *Nicotiana tabacum* [16]. This lack of an endogenous glycosyl transferase in this specific subcellular compartment may be the reason why no diglucose-conjugates were formed. Expression of the metabolic pathway to artemisinic acid in *Physcomitrella patens* resulted in the accumulation of nongylcosylated artemisinic acid in the apoplast [18]. Therefore, if glycosylation to an acid-group appears, a promising strategy would be to target a different subcellular localization or to change the expression system.

### 4.3. Acid-Induced Rearrangement

Acidic conditions can arise from unbuffered yeast media, as yeast cells rapidly decrease the neutral pH of the culture medium to a pH of ca. 3 [10,20]. Also, acid-induced rearrangement can occur in solid-phase microextraction gas chromatography (SPME-GC) analysis which was shown for the analysis of germacrene D [32] or when acidified solvents are used in high performance liquid chromatography (HPLC) analysis [10]. Recently, it was shown that acid-induced rearrangement may also play a role in plant expression systems and may form rearrangement product combinations with Michael-type adducts [13]. To overcome acid-induced rearrangement in yeast culture, yeast cultures can be buffered [7,39]. The use of MOPS buffer has proven to be better compatible with yeast growth than HEPES buffer [7,39]. If an acidified HPLC system is necessary, run times can be reduced and when preparative HPLC runs are performed, fractions can be collected in neutral buffer [10,39]. The right choice of SPME fibers reduces the acid-induced conversion of enzyme products in SPME-GC-MS analysis [32].

Acidic conditions can induce several other rearrangement reactions. The comparison of kunzeaol production in buffered and unbuffered yeast cultures showed a wide range of murolene and cadinene rearrangement products [40]. Andersen et al. (2015) showed up to 12 acid-induced rearrangement products of germacrene D depending on SPME-GC inlet temperature and fiber material [32].

### 4.4. Heat-Induced Rearrangement

Enzyme products of sesquiterpene synthases are usually analyzed by GC-MS. A high temperature in the GC inlet causes for instance the backbone of germacrene to rearrange to elemene. Reducing the inlet temperature of the gas chromatography system can overcome the problem with artificial heat-induced rearrangement products [32].

## 5. How Can Unspecific Enzyme Products Be Identified?

### 5.1. S-Conjugation

Generally, it is advisable to use nontargeted metabolomics approaches that can lead to the detection of enzyme products and their rearrangement products that were previously not anticipated [7,28]. However, there are several hints that can make the analysis of HPLC, LC-MS or GC-MS runs of putative sesquiterpenoid enzyme products easier. The S-conjugation to cysteine (a) increases the molecular weight from [M(STL)] to [M(STL) + 121], and the conjugation to glutathione increases it from [M(STL)] to [M(STL) + 307] [7,13,14]. This mass shift is a good indication in LC-MS analysis. Cysteine adducts and glutathione adducts elute earlier than the unconjugated STL in reverse-phase HPLC systems. Usually the cysteine and glutathione conjugates of the same STL appear as a peak tandem, with the cysteine adduct being more dominant and eluting slightly earlier [7,13,14]. A good indication for these adducts is also the isotope pattern of the mass peak that indicates the presence of the single sulfur atom from the thiol group of cysteine being introduced into the molecule [13,14]. If the STL is available as a reference compound, synthetic conjugates can easily be produced as reference compounds via Michael-type addition by incubation with cysteine- and glutathione at room temperature [7].

### 5.2. O-Conjugation

O-conjugation to a sesquiterpene will result in earlier elution from the reversed-phase column and a mass shift from [M(ST)] to [M(ST) + 324] (the mass of two glucose units minus the mass of two water units) [28].

### 5.3. Acid-Induced Rearrangement

Acid-induced rearrangement reactions can result in many products that are very similar to the enzyme product itself. In the case of a yeast expression system, culture and extraction conditions can be altered to differentiate between specific and unspecific enzyme products. For instance, Nguyen et al. (2010) could show an increase of the specific enzyme product germacrene A acid (**10**) and a decrease of the rearrangement products costic acids and ilicic acid (**11**–**14**) by comparing extracts from buffered versus unbuffered yeast cultures on a GC-MS and a LC-MS system [10]. Co-elution of acid-induced isomers on GC-MS can be overcome by the use of chiral columns which allow the separation of α-costic acid (**11**) and β-costic acid (**12**) [10].

The acid-induced transannular cyclization of germacranolide ST has been shown to lead to multiple rearrangement products that differ only in the position of one double bond [9,10,13]. These isomers are very difficult to separate from each other as they often co-elute or have very similar retention times on reversed-phase HPLC systems [9,10,13]. Furthermore, they have the same exact mass and strikingly similar MS-MS fragmentation patterns [9,10,13]. The recently reported rearrangement products from the expression of the pathway to germacrene A acid (**10**) in *Nicotiana benthamiana* indicate that acid-induced rearrangement may also appear in the subcellular environment of plant cells used for heterologous expression [13]. The combination of acid-induced rearrangement products with Michael-type reaction can lead to a very complex matrix of rearrangement products, that may be nearly impossible to separate [13]. To disentangle this matrix of products the production of the cysteine and glutathione adducts of reference compounds that match the presumed acid-induced rearrangement products was reported as a solution [13].

If the acid-induced rearrangement is followed by the introduction of a water molecule the resulting rearrangement is easier to spot as the additional hydroxy-group results in an earlier elution from the reversed-phase chromatography system and the molecular weight increases from [M(ST)] to [M(ST) + 18]. This was observed for the rearrangement from germacrene A acid (**10**) to ilicic acid (**14**) and from 8β-hydroxygermacrene A acid to 8β-hydroxyilicic acid [9,10]. Interestingly, costunolide (**1**) and its derivatives, germacranolides with similar structure, appear more stable in acidic conditions and do not show acid-induced rearrangements [9,14].

### 5.4. Heat-Induced Rearrangement

Similar to acid-induced rearrangements, heat-induced rearrangement does not change the total mass and only slightly changes the MS-MS pattern for the conversion from germacrene A (**15**) to β-elemene (**16**). However, germacrene A (**15**) elutes later (at higher temperatures) from the GC-MS column than its heat-induced rearrangement product β-elemene (**16**) [11].

## 6. How are Unspecific Reactions Prevented in the Natural Situation?

All the above-mentioned unspecific reactions could theoretically occur in the natural plant except for the heat-induced Cope rearrangement. Why are these reactions rarely observed in nature, where plants can accumulate highly reactive enzyme products in high concentrations in a way that provides protection without poisoning the host plant cell? In the case of STL, several biosynthetic enzymes that have so far been characterized are naturally located in secretory cells of glandular trichomes [8,11,13] or secretory ducts [29]. The upregulation of STL enzyme expression has been shown to be tightly linked to trichome development [8,11,13], which would allow an efficient production of STL before they are transported out of the cell.

## 7. Outlook

For researchers engaging in the reconstruction of metabolic pathways, it is crucial to have a perception about the unspecific reactions the presumed enzyme product may undergo. Here, a systematic approach to categorize and explain the most frequent unspecific reactions for sesquiterpenes is presented. Expanding this approach to other classes of natural compounds such as mono-, di- and triterpenes as well as polyphenols and alkaloids could create a store of knowledge to better plan and interpret the reconstruction of the metabolic pathways of plant specialized metabolites.

## Figures and Tables

**Figure 1 molecules-25-01935-f001:**
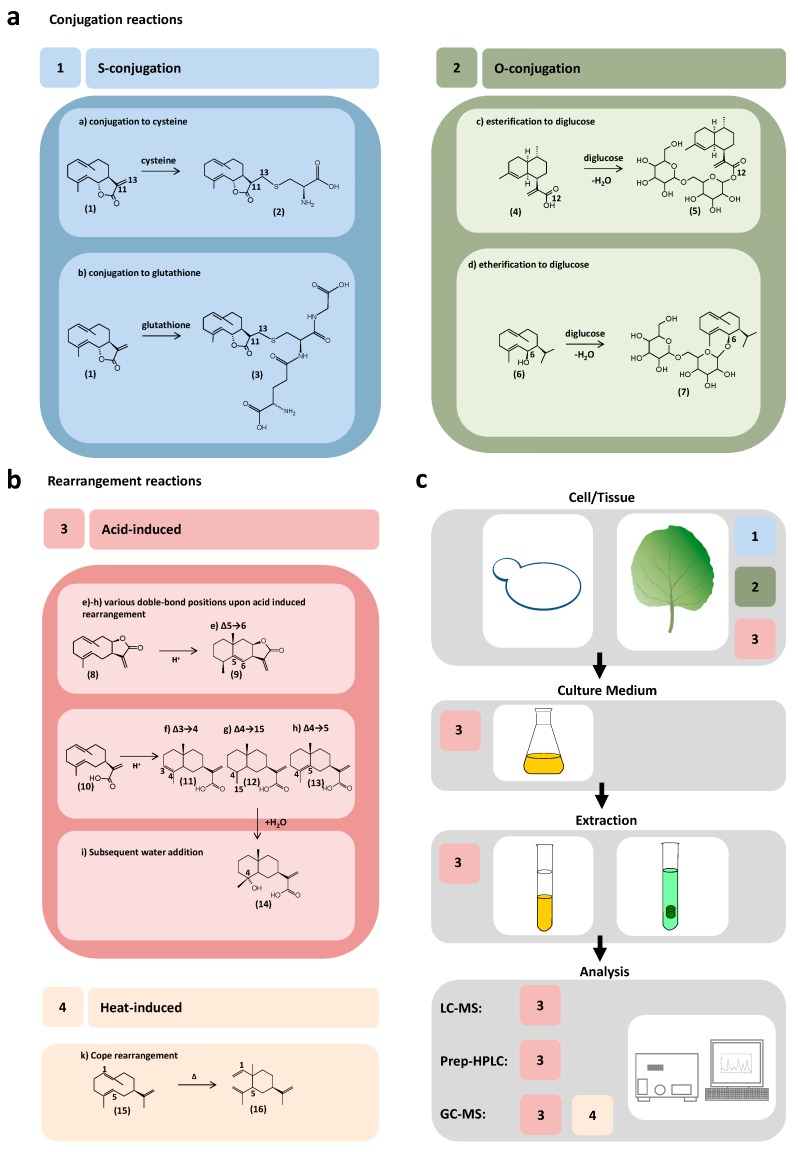
Unspecific rearrangement and conjugation reactions of sesquiterpenoids in heterologous expression systems. (**a**) Conjugation reactions; (**b**) Rearrangement reactions; (**c**) Unspecific reactions in the workflow of enzyme characterization.

**Table 1 molecules-25-01935-t001:** Classification and avoidance of unspecific reactions.

Group	Type	Combinations	Modification	Mass	Functional Group	Reaction Type	Cause	Avoiding Strategies	References
1. S-conjugation	a	f–i	STL-GSH	+307	α-methylene-γ-lactone	Michael-type addition, or GST reaction	Plant cell detoxification	Targeting different subcellular localizations	[7,8,13]
	b	f–i	STL-Cys	+121					[7,8,13]
2. O-conjugation	c		STLOH-OGlc_2_	+324	acid-group	Esterification or	Plant cell detoxification, presumably enzymatic	Targeting different subcellular localizations	[18,28]
					hydroxy- group	etherification			[29]
	d		STL-OGlc_2_						
3. Acid	e	a–b	STL**, Δ3→4	±0	1,5-diene (Germa- crene)	Acid-induced rearrangement	pH in culture media, cells, chromatography	Buffering of yeast media, pH control of chromatography system, choice of SPME fibers	[9,10,13]
	f	a–b	STL**, Δ4→15						[9,10,13]
	g	a–b	STL**, Δ4→5						[9,10,13]
	h	a–b	STL**, Δ5→6						[13]
	i		STL**-OH	+18					[9,10]
4. Heat	k		STL*	±0	1,5-diene (Germa- crene)	Cope rearrangement (heat-induced)	GC-MS Analysis	Reduction of heat in GC-MS inlet	[11,32,33]

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
