# Peer review of "Traps and Pitfalls—Unspecific Reactions in Metabolic Engineering of Sesquiterpenoid Pathways"

_molecules, 2020, doi:10.3390/molecules25081935_

Round 1

Reviewer 1 Report

Review of Traps and pitfalls - Unspecific reactions in metabolic engineering of sesquiterpenoid pathways by Maximilian Frey.

This mini-review reports the rearrangement reactions described when heterologous expressions of sesquiterpenes were performed. But also it presents the clues to avoid and to identify them. This article is a really useful toolbox for researchers in the field of sesquiterpenes but as a consequence is restricted to a small part of people working on isoprenoids.

The author doesn't mentioned the hexose/malonyl conjugates found in heterologous expression of artemisinin in tobacco described by Zhang et al. (2011). Any reason for that?

Minor comments

  1. In the abstract part:

Lanes 10 and 11, the author seems to include bacteria among the eukaryotic cells.

Lanes 14-16, the same sentence seems to be duplicated with other words.

Lane 25, "pant" instead of "plant".

  1. Lanes 50-52, the same sentence is duplicated.
  2. Lane 88, paragraph 3, there is no 3.2 section, so the 3.1 section is not necessary, or the 3.2 part is lacking?
  3. Lane 95, generalities about S- and O-conjugates in the beginning of the 3.1.1. paragraph should be removed.
  4. Lane 103, in the article by Liu et al., 2011 (7) there is no reference to non-enzymatic Michael-type modification.
  5. Lane 116, " The cysteine conjugates are interpreted as be breakdown products of the cysteine adducts or result", the second "cysteine" is probably "glutathione"
  6. Lane 134, in the article by Liu et al., 2011 (7) there is no reference to the decrease of the yeast culture medium pH.

Author Response

Reviewer 1

  • The reviewer´s encouraging remarks are highly appreciated. All minor comments have been addressed.

Review of Traps and pitfalls - Unspecific reactions in metabolic engineering of sesquiterpenoid pathways by Maximilian Frey.

This mini-review reports the rearrangement reactions described when heterologous expressions of sesquiterpenes were performed. But also it presents the clues to avoid and to identify them. This article is a really useful toolbox for researchers in the field of sesquiterpenes but as a consequence is restricted to a small part of people working on isoprenoids.

  • It is very encouraging that the article is considered useful to the researchers in the field of sesquiterpenoids.

  • Unquestionably for the broader community working on plant metabolic pathways in general the principle of systematically categorizing unspecific reactions together with a guide how to avoid them is interesting as well.

The author doesn't mentioned the hexose/malonyl conjugates found in heterologous expression of artemisinin in tobacco described by Zhang et al. (2011). Any reason for that?

  • Assuming that the reviewer refers to “The production of artemisinin precursors in tobacco” by (Zhang et al., 2011) this reference was taken into consideration. I could not find “hexose/malonyl conjugates found in heterologous expression of artemisinin in tobacco” being mentioned. Zhang et al. (2011) speculate about the involvement of an endogenous tobacco dehydrogenase, but this is discussed as one of several possible explanations for their findings.

Minor comments

  1. In the abstract part:

Lanes 10 and 11, the author seems to include bacteria among the eukaryotic cells.

  • The sentence was rephrased for clarification.

“The characterization of plant enzymes by expression in prokaryotic and eukaryotic (yeast and plants) heterologous hosts has widely been used […] Yeast and plant systems provide the cellular environment of a eukaryotic cell…”

Lanes 14-16, the same sentence seems to be duplicated with other words.

  • A sentence was removed to avoid redundancy.

“However, [removed redundant sentence] in many cases the detected compounds are not the direct enzyme products but are caused by unspecific subsequent reactions.”

Lane 25, "pant" instead of "plant".

  • The spelling error was corrected.

„…other classes of plant secondary…”

  1. Lanes 50-52, the same sentence is duplicated.
  • A sentence was removed to avoid redundancy.

 “… observed and several various combinations of these can occur. These unspecific reactions are numbered from (a) to (k). [removed redundant sentence]. Irradiation…”

  1. Lane 88, paragraph 3, there is no 3.2 section, so the 3.1 section is not necessary, or the 3.2 part is lacking?
  • The numbering has been corrected.
  1. Lane 95, generalities about S- and O-conjugates in the beginning of the 3.1.1. paragraph should be removed.
  • Now Lane 103, Paragraph was shortened

“[removed] S-conjugation to the thiol group of cysteine or O-conjugation to the hemiacetal group of glucose are mainly interpreted…”

  1. Lane 103, in the article by Liu et al., 2011 (7) there is no reference to non-enzymatic Michael-type modification.
  • Now Lane 111, Liu et al. (2011) have shown that the non-enzymatic reaction of glutathione to costunolide is possible (which is in line with many studies on the chemistry of α-methylene-γ-lactone sesquiterpene lactones). On page 4 in the paragraph “Identification of costunolide conjugates” the authors describe the findings as follows: “When costunolide and glutathione were incubated without GST enzyme, the same costunolide-glutathione conjugate was formed indicating that the conjugation of costunolide and glutathione can occur spontaneously " (Liu et al., 2011)
  1. Lane 116, " The cysteine conjugates are interpreted as be breakdown products of the cysteine adducts or result", the second "cysteine" is probably "glutathione"
  • Now Lane 126, The word “cysteine” has been replaced by “glutathione”.
  1. Lane 134, in the article by Liu et al., 2011 (7) there is no reference to the decrease of the yeast culture medium pH.
  • Now Lane 142, The citation at this position was removed.

References:

Liu, Q., Majdi, M., Cankar, K., Goedbloed, M., Charnikhova, T., Verstappen, F. W. A., de Vos, R. C. H., Beekwilder, J., van der Krol, S., & Bouwmeester, H. J. (2011). Reconstitution of the costunolide biosynthetic pathway in yeast and Nicotiana benthamiana. PLoS ONE, 6(8). https://doi.org/10.1371/journal.pone.0023255

Zhang, Y., Nowak, G., Reed, D. W., & Covello, P. S. (2011). The production of artemisinin precursors in tobacco. Plant Biotechnology Journal, 9(4), 445–454. https://doi.org/10.1111/j.1467-7652.2010.00556.x

Reviewer 2 Report

The manuscript molecules-778182 an interesting overview on plant enzymes of sesquiterpenoid pathways.

The manuscript is well organized and the bibliographic research is sufficient to a mini-review. The discussion is sufficiently argued. I consider that this manuscript is appropriate for publication in Molecules without revisions.

Author Response

Reviewer 2

  • The reviewer´s encouraging remarks are highly appreciated.

The manuscript molecules-778182 an interesting overview on plant enzymes of sesquiterpenoid pathways.

The manuscript is well organized and the bibliographic research is sufficient to a mini-review. The discussion is sufficiently argued. I consider that this manuscript is appropriate for publication in Molecules without revisions.

Reviewer 3 Report

This mini-review summarizes unspecific reactions observed when plant natural product biosynthesis is reconstituted in heterologous hosts and the strategies to avoid these reactions. This paper will be of interest to the researchers working on the biosynthesis and biosynthetic engineering of plant metabolites. Thus, I support the publication of the manuscript in the journal after the author addresses the following points.

  1. In the Introduction, the author states that “In a second step the intermediate is modified by cytochrome P450 enzymes that introduces oxygen into the core backbone”, but this is not always true; other enzymes than P450s can also be involved in the tailoring of terpene skeletons. Please rephrase accordingly.
  2. Provide relevant atom numbers in Figure 1 for easier understanding.
  3. Figure 1b – Regarding the formation of 14, I believe it is more likely that a water molecule is used to neutralize the carbocationic species to give 14.
  4. It would be better if the author could discuss how plants that originally produce a certain natural product avoid the formation of conjugated or rearranged products.

Author Response

Reviewer 3

  • The reviewer´s encouraging remarks are highly appreciated. All points have been addressed.

This mini-review summarizes unspecific reactions observed when plant natural product biosynthesis is reconstituted in heterologous hosts and the strategies to avoid these reactions. This paper will be of interest to the researchers working on the biosynthesis and biosynthetic engineering of plant metabolites. Thus, I support the publication of the manuscript in the journal after the author addresses the following points.

1. In the Introduction, the author states that “In a second step the intermediate is modified by cytochrome P450 enzymes that introduces oxygen into the core backbone”, but this is not always true; other enzymes than P450s can also be involved in the tailoring of terpene skeletons. Please rephrase accordingly.

  • Lane 39, The sentence has been rephrased to point out that the introduction of oxygen is not exclusively mediated by P450 enzymes.

“In a second step the intermediate is often modified by cytochrome P450 enzymes that introduces oxygen into the core backbone.”

2. Provide relevant atom numbers in Figure 1 for easier understanding.

  • Figure 1, Lane 70, Relevant atom numbers are now provided.

3. Figure 1b – Regarding the formation of 14, I believe it is more likely that a water molecule is used to neutralize the carbocationic species to give 14.

  • Lane 88, This is a good explanation for the observed reaction and has been included in the text.

“The subsequent introduction of water (i) resulting in ilicic acid (14) has also been reported 9,10, likely neutralizing a carbocationic intermediate”

4. It would be better if the author could discuss how plants that originally produce a certain natural product avoid the formation of conjugated or rearranged products.

  • Lanes 215-223, A short discussion concerning this question has been included.